# IME: Integrating Multi-curvature Shared and Specific Embedding for Temporal Knowledge Graph Completion

Submission Id: 295*

## ABSTRACT

Temporal Knowledge Graphs (TKGs) incorporate a temporal dimension, allowing for a precise capture of the evolution of knowledge and reflecting the dynamic nature of the real world. Typically, TKGs contain complex geometric structures, with various geometric structures interwoven. However, existing Temporal Knowledge Graph Completion (TKGC) methods either model TKGs in a single space or neglect the heterogeneity of different curvature spaces, thus constraining their capacity to capture these intricate geometric structures. In this paper, we propose a novel Integrating Multi-curvature shared and specific Embedding (IME) model for TKGC tasks. Concretely, IME models TKGs into multi-curvature spaces, including hyperspherical, hyperbolic, and Euclidean spaces. Subsequently, IME incorporates two key properties, namely *space-shared property* and *space-specific property*. The space-shared property facilitates the learning of commonalities across different curvature spaces and alleviates the spatial gap caused by the heterogeneous nature of multi-curvature spaces, while the space-specific property captures characteristic features. Meanwhile, IME proposes an Adjustable Multi-curvature Pooling (AMP) approach to effectively retain important information. Furthermore, IME innovatively designs similarity, difference, and structure loss functions to attain the stated objective. Experimental results clearly demonstrate the superior performance of IME over existing state-of-the-art TKGC models.

## KEYWORDS

Temporal Knowledge Graph, Knowledge Graph Completion, Multi-curvature Embeddings, Adjustable Pooling

## 1 INTRODUCTION

Knowledge Graphs (KGs) are structured collections of entities and relations, providing a semantic representation of knowledge. They serve as a powerful tool for organizing and representing real-world information in a way that machines can comprehend. Typically, knowledge in KGs is represented as *triplets*, where each node is represented as an entity, and the directed edge between nodes is denoted as a relation. For example, given one *triplet (Albert Einstein, born_in, Germany)*, *Albert Einstein* and *Germany* are the head and tail entities, and *born_in* means the relation between the head

*WWW '24, May 13-17, 2024, Singapore*
© 2024 Association for Computing Machinery.
ACM ISBN 978-1-4503-XXXX-X/18/06...$15.00
https://doi.org/XXXXXXX.XXXXXXX

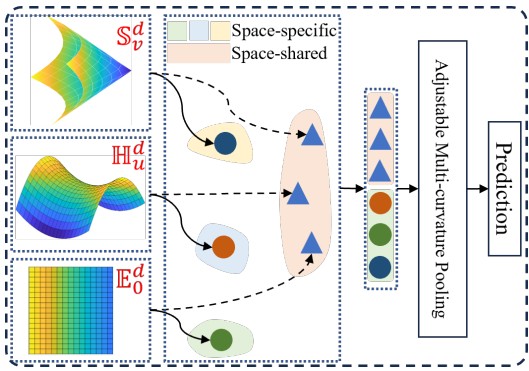

**Figure 1: A brief description of IME. Learning multi-curvature representations through *space-shared* and *space-specific* properties. These features are later utilized for subsequent predictions by the adjustable multi-curvature pooling.**

and tail entities. KGs find applications in a wide array of domains, including recommendation systems [19], information retrieval [8], and semantic search [14]. They enable machines to reason about entities and their relations, uncover patterns, and make informed decisions based on the structured knowledge they encapsulate.

Acknowledging the ever-changing nature of information, Temporal Knowledge Graphs (TKGs) have arisen as a natural extension of traditional KGs. In contrast to their static counterparts, TKGs introduce the temporal dimension, enabling us to track the evolution of knowledge over time. Specifically, TKGs aim to incorporate temporal attributes with triplets for *quadruplets*: *(Albert Einstein, born_in, Germany, 1879-03-14)*, with *1879-03-14* serving as the timestamp. Therefore, the temporal dimension allows for a systematic depiction of the trends and changes in events, thereby facilitating more context-aware and precise knowledge representation. **TKGs directly address the dynamic nature of the web, enabling the understanding of "Semantics and Knowledge" of the Web**, making it highly pertinent to the theme of this conference.

Despite the presence of TKGs like ICEWS [4] and GDELT [22], which encompass millions or even billions of quadruplets, the ongoing evolution of knowledge driven by natural events leaves these TKGs far from being comprehensive. The incompleteness of TKGs poses a substantial hindrance to the efficiency of knowledge-driven systems, underscoring the critical significance of "Temporal Knowledge Graph Completion (TKGC)" as an essential undertaking. The goal of the TKGC task is to enhance the completeness and accuracy of TKGs by predicting missing relations, entities, or temporal attributes that change over time within the TKGs.

TKGs often encompass complex geometric structures, and effectively modeling them is crucial for unlocking the full potential

of TKGC in real-world applications. As depicted in [5, 37], various curvature spaces yield diverse impacts when used to model distinct geometric structures. Specifically, hyperspherical space excels in capturing ring structures, hyperbolic space is highly effective in representing hierarchical arrangements, and Euclidean space proves invaluable for describing chain-like structures. Nonetheless, the majority of TKGC methods typically model TKGs within a singular space, posing a challenge in effectively capturing the intricate geometric structures inherent in TKGs.

The challenge of how to effectively integrate information from different curvature spaces subsequently needs to be addressed. Current TKGC methods [16, 45] typically overlook the spatial gap among different curvature spaces. Despite significant advancements, the spatial gap remains a substantial constraint on expressive capacities.

The last challenge is the feature fusion issue. Existing methods [34, 42] predominantly focus on developing sophisticated fusion mechanisms, causing a high computational complexity. Despite the effectiveness of pooling approaches like average pooling and max pooling in reducing computational complexity, their utilization of fixed pooling strategies presents a challenge in preserving important information.

This paper proposes a novel Integrating Multi-curvature shared and specific Embedding (IME) model to address the above challenges. As shown in Figure 1, IME simultaneously models TKGs in hyperspherical, hyperbolic, and Euclidean spaces, introducing the quadruplet distributor [34] within each space to facilitate the aggregation and distribution of information among entities, relations, and timestamps. In addition, IME acquires two distinct properties for each space, encompassing both *space-shared* and *space-specific* properties. The space-shared property aids in mitigating the space gap by capturing shared information among entities, relations, and timestamps across various curvature spaces. Conversely, the space-specific property excels at fully capturing the complementary information exclusive to each curvature space. Finally, an Adjustable Multi-curvature Pooling (AMP) approach is proposed, which can learn appropriate pooling weights to get a superior pooling strategy, ultimately improving the effective retention of important information. We utilize AMP to aggregate space-shared and -specific representations of entities, relations, and timestamps to get a joint vector for downstream predictions.

The main contributions of this paper are summarized as follows:

- This paper designs a novel Multi-curvature Space-Shared and -Specific Embedding (IME) model for TKGC tasks, which learns two key properties, namely *space-shared* property and *space-specific* property. Specifically, space-shared property learns the commonalities across distinct curvature spaces and mitigates spatial gaps among them, while space-specific property captures characteristic features;
- This paper proposes an adjustable multi-curvature pooling module, designed to attain a superior pooling strategy through training for the effective retention of important information;
- To the best of our knowledge, we are the first to introduce the concept of structure loss into TKGC tasks, ensuring the

structural similarity of quadruplets across various curvature spaces;
- Experimental results on several widely used datasets demonstrate that IME achieves competitive performance compared to state-of-the-art TKGC methods.

## 2 RELATED WORK

In this section, we provide an overview of KGC methods from two perspectives [36]: *Euclidean embedding-based methods* and *Non-Euclidean embedding-based methods*.

### 2.1 Euclidean Embedding-based Methods

Euclidean embedding-based KGC methods typically model the KGs in the Euclidean space. Depending on the types of knowledge, we can categorize them into *static knowledge graph completion* for triplets and *temporal knowledge graph completion* for quadruplets.

**Static knowledge graph completion (SKGC)** focuses on SKGs where the information about entities and relations remains unchanged over time. The task of SKGC methods aims to predict missing triplets (e.g., relations between entities) based on known information. Several popular SKGC methods include McRL [33], TDN [35], and ConvE [10].

*Translation-based methods* take the relation as a translation from the head entity to the tail entity, such as TransE [3] and RotatE [29]. RotatE regards the relation as a rotation from the head entity to the tail entity in the complex space. Based on TransE, TransR [23] learns a unified mapping matrix to model the entities and relations into a common space. SimplE [18] improves upon the complex Canonical Polyadic (CP) decomposition [17] by enabling the interdependent learning of the two embeddings for each entity within the complex space. Furthermore, BoxE [1] introduces the box embedding method as a means to model the uncertainty and diversity inherent in knowledge.

*Semantic matching-based methods* employ a similarity-based scoring function to evaluate the probabilities of triplets, such as DistMult [41] and McRL [33]. DistMult employs matrix multiplication to model the interaction between the entity and relation. ComplEx [30] operates within the complex space to calculate the score of the triplet. CapsE [32] introduces the capsule network to capture the hierarchical relations and semantic information among entities. TuckER [2] explores Tucker decomposition into the SKGC task. In addition, McRL [33] captures the complex conceptual information hidden in triplets to acquire accurate representations of entities and relations.

*Convolutional neural network-based methods* explore the use of CNN to capture the inherent correlations within triplets. ConvE [10] first employs the CNN into the SKGC task. R-GCN [26] explores the graph neural network to update entity embeddings. Moreover, TDN [35] creatively designs the triplet distributor to facilitate the information transmission between entities and relations.

**Temporal knowledge graph completion (TKGC)** refers to the prediction of unknown quadruplets in TKGs based on known information, including entities, relations, and timestamps. Some classic TKGC methods contain ChronoR [25], TeLM [38] and BoxTE [24].

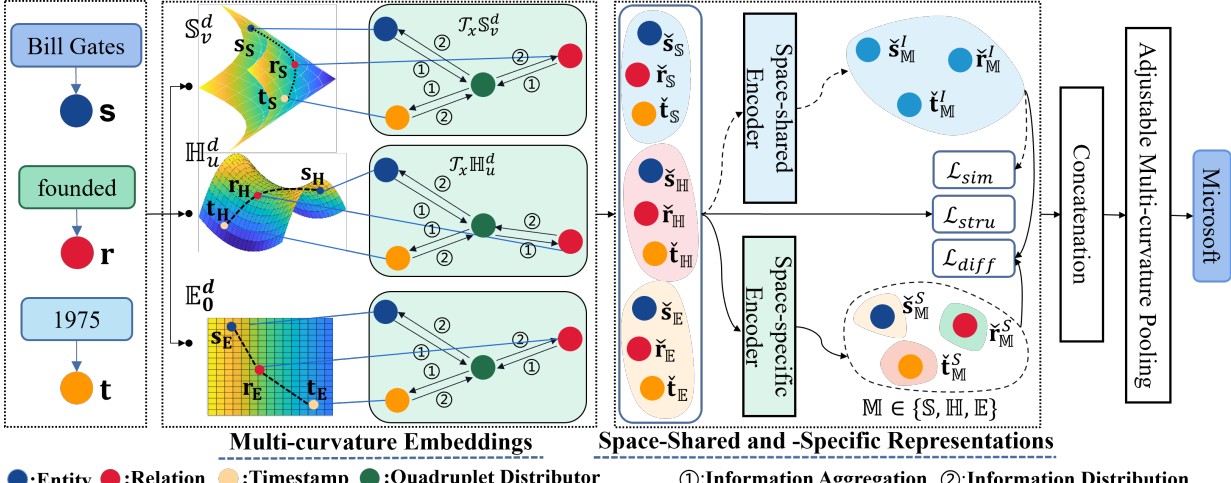

**Figure 2: The framework of IME. Specifically, IME models the query *(Donald Trump, president of, ?, 2018)* in multi-curvature spaces through information aggregation and information distribution. Subsequently, IME explores *space-shared* and *space-specific* properties to learn the commonalities and characteristics across different curvature spaces, effectively reducing spatial gaps among them. Finally, these identified features are employed for adjustable multi-curvature pooling in subsequent predictions.**

TTransE [21] models the pair of the relation and timestamp as the translation between the head entity and the tail entity. TA-TransE and TA-DistMult [12] integrate timestamps into entities using recurrent neural networks to capture the dynamic evolution of entities. Building upon RotatE, ChronoR represents relation-timestamp pairs as rotations from the head entity to the tail entity. Similarly, TuckERTNT [28] extends the 3rd-order tensor to the 4th-order to model quadruplets. More recently, BoxTE [24] has been introduced to enable more versatile and flexible knowledge representation.

HyTE [9] first explores the dynamic evolution of entities and relations by modeling entities and relations into the timestamp space. TeRo [40] models the temporal evolution of entities as a rotation in complex vector space, and handles time interval facts using dual complex embeddings for relations. TComplEx [20] is based on ComplEx, which expands the 3rd-order tensor into a 4th-order tensor to perform TKGC. DE-SimplE [13] designs the diachronic entity embedding function to capture the dynamic evolution of entities over time, subsequently employing SimplE for predicting missing items. ATiSE [39] decomposes timestamps into the trend, seasonal, and irregular components to capture the evolution of entities and relations over time. TeLM [38] employs multivector embeddings and a linear temporal regularizer to obtain entity and timestamp embeddings, respectively. EvoExplore [44] incorporates two critical factors for comprehending the evolution of TKGs: local structure describes the formation process of the graph structure in detail, and global structure reflects the dynamic topology of TKGs. BDME [42] leverages the interaction among entities, relations, and timestamps for coarse-grained embeddings and block decomposition for fine-grained embeddings. Particularly, QDN [34] extends the triplet distributor [35] into a quadruplet distributor and designs

the 4th-order tensor decomposition to facilitate the information interaction among entities, relations, and timestamps.

## 2.2 Non-Euclidean Embedding-based Methods

Non-Euclidean embedding-based methods typically embed KGs into non-Euclidean space, effectively capturing the complex geometric structure inherent to them. Some classic non-Euclidean embedding methods include ATTH [6], MuRMP [37], and BiQCap [45].

For **SKGC**, ATTH models the KG within the hyperbolic space to capture both hierarchical and logical patterns. BiQUE [15] utilizes biquaternions to incorporate multiple geometric transformations, including Euclidean rotation, which is valuable for modeling patterns like symmetry, and hyperbolic rotation, which proves effective in capturing hierarchical relations. MuRMP and GIE [5] simultaneously model the KG within multi-curvature spaces to capture the complex structure.

For **TKGC**, DyERNIE [16] embeds TKGs into multi-curvature spaces to explore the dynamic evolution guided by velocity vectors defined in the tangent space. BiQCap [45] simultaneously models each relation in Euclidean and hyperbolic spaces to represent hierarchical semantics and other relation patterns of TKGs.

## 3 PROBLEM DEFINITION

Temporal knowledge graph $\mathcal{G} = \{Q \mid \mathcal{E}, \mathcal{R}, \mathcal{T}\}$ is a collection of entity set $\mathcal{E}$, relation set $\mathcal{R}$ and timestamp set $\mathcal{T}$. Specifically, each quadruplet in $\mathcal{G}$ is denoted as $(\mathbf{s}, \mathbf{r}, \mathbf{o}, \mathbf{t}) \in Q$, where $\mathbf{s}, \mathbf{o} \in \mathcal{E}$ represent the head and tail entities, $\mathbf{r} \in \mathcal{R}$ denotes the relation and $\mathbf{t} \in \mathcal{T}$ is the timestamp. The primary objective of the TKGC task is to predict the missing tail entity when given a query $(\mathbf{s}, \mathbf{r}, ?, \mathbf{t})$, or the missing head entity when provided with a query $(?, \mathbf{r}, \mathbf{o}, \mathbf{t})$.

## 4 METHODOLOGY

In this section, we present a detailed description of IME, which can be segmented into three main stages: Multi-curvature Embeddings, Space-shared and -specific Representations, and Adjustable Multi-curvature Pooling. The whole framework is illustrated in Figure 2.

### 4.1 Multi-curvature Embeddings

TKGs typically encompass intricate geometric structures, including ring, hierarchical, and chain structures. Specifically, distinct geometric structures are characterized by differing modeling capacities across various geometric spaces. We simultaneously model TKGs in multi-curvature spaces to capture the complex structures.

Inspired by QDN [34], for each curvature space, we introduce the quadruplet distributor to facilitate the information aggregation and distribution among them. This is due to the fact that entities, relations, and timestamps within each curvature space typically exist in distinct semantic spaces, hindering the information transmission among them.

Given the entity, relation, timestamp, and the initial zero-tensor of the quadruplet distributor, denoted as $\mathbf{s}$, $\mathbf{r}$, $\mathbf{t}$, and $\mathbf{q}$, we operate the *information aggregation* and *information distribution*.

**Information Aggregation** dynamically aggregates the information of entities, relations, and timestamps into the quadruplet distributor through gating functions,

$$\begin{aligned}
\mathbf{s_{q1}} &= (\mathbf{s} - \mathbf{q}) \odot [\sigma(\mathbf{s} - \mathbf{q})] \\
\mathbf{r_{q1}} &= (\mathbf{r} - \mathbf{q}) \odot [\sigma(\mathbf{r} - \mathbf{q})] \\
\mathbf{t_{q1}} &= (\mathbf{t} - \mathbf{q}) \odot [\sigma(\mathbf{t} - \mathbf{q})],
\end{aligned} \tag{1}$$

where $\sigma$ represents the sigmoid activation function; $\odot$ is the element-wise multiplication.

Subsequently, we employ the residual network to aggregate the information of the entity, relation, and timestamp into the quadruplet distributor,

$$\check{\mathbf{q}} = \mathbf{q} \oplus \mathbf{s_{q1}} \oplus \mathbf{r_{q1}} \oplus \mathbf{t_{q1}}. \tag{2}$$

**Information Distribution** distributes the above aggregated quadruplet distributor $\check{\mathbf{q}}$ to the entity $\mathbf{s}$, relation $\mathbf{r}$ and timestamp $\mathbf{t}$ through gating functions,

$$\begin{aligned}
\mathbf{s_{q2}} &= (\mathbf{s} - \check{\mathbf{q}}) \odot [\sigma(\mathbf{s} - \check{\mathbf{q}})] \\
\mathbf{r_{q2}} &= (\mathbf{r} - \check{\mathbf{q}}) \odot [\sigma(\mathbf{r} - \check{\mathbf{q}})] \\
\mathbf{t_{q2}} &= (\mathbf{t} - \check{\mathbf{q}}) \odot [\sigma(\mathbf{t} - \check{\mathbf{q}})].
\end{aligned} \tag{3}$$

Finally, we distribute the information of quadruplet distributor into the entities, relations, and timestamps,

$$\begin{aligned}
\check{\mathbf{s}} &= \mathbf{s} \oplus \mathbf{s_{q2}} \\
\check{\mathbf{r}} &= \mathbf{r} \oplus \mathbf{r_{q2}} \\
\check{\mathbf{t}} &= \mathbf{t} \oplus \mathbf{t_{q2}}.
\end{aligned} \tag{4}$$

Through the above information aggregation and information distribution process, we can obtain updated representations of entities, relations, and timestamps $\check{\mathbf{s}}$, $\check{\mathbf{r}}$ and $\check{\mathbf{t}}$.

Similarly, we operate the above information aggregation and information distribution in multi-curvature spaces, including Euclidean, hyperbolic, and hyperspherical spaces. For each entity $\mathbf{s}$, relation $\mathbf{r}$, and timestamp $\mathbf{t}$, we can obtain their features in three

curvature spaces, namely $\check{\mathbf{s}}_{\mathbb{M}}$, $\check{\mathbf{r}}_{\mathbb{M}}$, and $\check{\mathbf{t}}_{\mathbb{M}}$ ($\mathbb{M} \in \{\mathbb{S}, \mathbb{H}, \mathbb{E}\}$). Thus, we obtain nine features.

### 4.2 Space-Shared and -Specific Representations

In order to facilitate the learning of commonalities across different curvature spaces, and comprehensively capture the characteristic features unique to each curvature space, we employ encoding functions to capture both *space-shared* and *space-specific* properties. Given the updated representations $\check{\mathbf{h}}_{\mathbb{M}}$ ($\mathbf{h} \in \{\mathbf{s}, \mathbf{r}, \mathbf{t}\}$, $\mathbb{M} \in \{\mathbb{S}, \mathbb{H}, \mathbb{E}\}$) of the entity, relation, and timestamp for different curvature spaces, we explore the gate attention mechanism to achieve the encoding functions.

**Space-shared** property focuses on recognizing commonalities across various curvature spaces to reduce spatial gaps among them. Specifically, it shares the parameters $\mathbf{W}_I$ in encoding function $E_I(\cdot)$ to obtain the space-shared representations. The encoding process can be denoted as,

$$\begin{aligned}
E_I(\check{\mathbf{h}}_{\mathbb{S}}) &= \check{\mathbf{h}}_{\mathbb{S}} \odot \sigma(\mathbf{W}_I \odot [\check{\mathbf{h}}_{\mathbb{S}}, \check{\mathbf{h}}_{\mathbb{H}}, \check{\mathbf{h}}_{\mathbb{E}}]) \\
E_I(\check{\mathbf{h}}_{\mathbb{H}}) &= \check{\mathbf{h}}_{\mathbb{H}} \odot \sigma(\mathbf{W}_I \odot [\check{\mathbf{h}}_{\mathbb{S}}, \check{\mathbf{h}}_{\mathbb{H}}, \check{\mathbf{r}}_{\mathbb{E}}]) \\
E_I(\check{\mathbf{h}}_{\mathbb{E}}) &= \check{\mathbf{h}}_{\mathbb{E}} \odot \sigma(\mathbf{W}_I \odot [\check{\mathbf{h}}_{\mathbb{S}}, \check{\mathbf{h}}_{\mathbb{H}}, \check{\mathbf{h}}_{\mathbb{E}}]),
\end{aligned} \tag{5}$$

where $\mathbf{W}_I$ is the shared parameter across all three curvature spaces, $[\cdot, \cdot, \cdot]$ represents the feature concatenation operation, $\odot$ is the element-wise multiplication, $\sigma$ denotes the Sigmoid function. Thus, we can generate nine space-shared representations $\mathbf{h}_{\mathbb{M}}^I$ ($\mathbf{h} \in \{\mathbf{s}, \mathbf{r}, \mathbf{t}\}$, $\mathbb{M} \in \{\mathbb{S}, \mathbb{H}, \mathbb{E}\}$) through the encoding functions $E_I(\check{\mathbf{h}}_{\mathbb{M}})$.

**Space-specific** property comprehensively captures the characteristic features unique to each curvature space. Similarly, it employs the encoding function $E_S(\cdot)$ to obtain the space-specific representations,

$$\begin{aligned}
E_S(\check{\mathbf{h}}_{\mathbb{S}}) &= \check{\mathbf{h}}_{\mathbb{S}} \odot \sigma(\mathbf{W}_S^1 \odot [\check{\mathbf{h}}_{\mathbb{S}}, \check{\mathbf{h}}_{\mathbb{H}}, \check{\mathbf{h}}_{\mathbb{E}}]) \\
E_S(\check{\mathbf{h}}_{\mathbb{H}}) &= \check{\mathbf{h}}_{\mathbb{H}} \odot \sigma(\mathbf{W}_S^2 \odot [\check{\mathbf{h}}_{\mathbb{S}}, \check{\mathbf{h}}_{\mathbb{H}}, \check{\mathbf{h}}_{\mathbb{E}}]) \\
E_S(\check{\mathbf{h}}_{\mathbb{E}}) &= \check{\mathbf{h}}_{\mathbb{E}} \odot \sigma(\mathbf{W}_S^3 \odot [\check{\mathbf{h}}_{\mathbb{S}}, \check{\mathbf{h}}_{\mathbb{H}}, \check{\mathbf{h}}_{\mathbb{E}}]),
\end{aligned} \tag{6}$$

where $\mathbf{W}_S^1$, $\mathbf{W}_S^2$, and $\mathbf{W}_S^3$ are the specific parameters unique to each curvature space. Similar to the space-shared property, we can generate nine space-specific representations $\mathbf{h}_{\mathbb{M}}^S$ ($\mathbf{h} \in \{\mathbf{s}, \mathbf{r}, \mathbf{t}\}$, $\mathbb{M} \in \{\mathbb{S}, \mathbb{H}, \mathbb{E}\}$) through the encoding functions $E_S(\check{\mathbf{h}}_{\mathbb{M}})$.

Through the above encoding functions $E_I(\cdot)$ and $E_S(\cdot)$, we can generate eighteen space-shared and -specific vectors $\mathbf{h}_{\mathbb{S}/\mathbb{H}/\mathbb{E}}^{I/S}$ ($\mathbf{h} \in \{\mathbf{s}, \mathbf{r}, \mathbf{t}\}$).

### 4.3 Adjustable Multi-curvature Pooling

After obtaining the space-shared and -specific representations of entities, relations, and timestamps, the pooling approach is employed to aggregate them into a joint vector for downstream predictions. We first introduce two simple pooling approaches: average pooling and max pooling. Then we introduce the proposed Adjustable Multi-curvature Pooling (AMP) approach.

As shown in Figure 3, for $n$ input features $\mathbf{X} = \{\mathbf{x}_1, \mathbf{x}_2, \cdots, \mathbf{x}_n\}$, $\mathbf{x}_i \in \mathcal{R}^{d_x}$, we sort each dimension of $n$ features to obtain the sorted features $\mathbf{M} = \{\mathbf{max}_1, \mathbf{max}_2, \cdots, \mathbf{max}_n\}$, $\mathbf{max}_i \in \mathcal{R}^{d_x}$. To get pooling feature $\mathbf{x}_p \in \mathcal{R}^{d_x}$, the pooling weights $\Psi = \{\psi_1, \psi_2, \cdots, \psi_n\}$, $\psi_i \in \mathcal{R}^1$ are used to perform a weighted sum over $\mathbf{M}$,

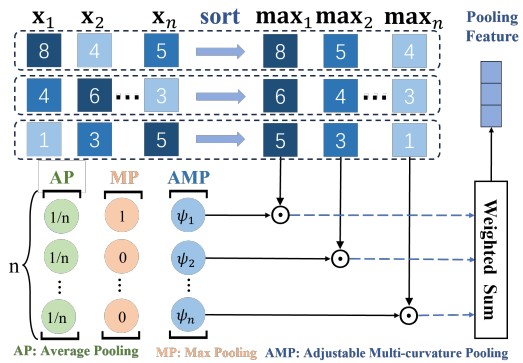

**Figure 3: Comparison of different pooling approaches.**

$$\mathbf{x}_p = \sum_{i=1}^{n} \psi_i \cdot \mathbf{max}_i. \tag{7}$$

**Average Pooling** sets all pooling weights $\psi_i$ to $\frac{1}{n}$ to get pooling feature $\mathbf{x}_{ap} \in \mathcal{R}^{d_x}$,

$$\mathbf{x}_{ap} = \sum_{i=1}^{n} \frac{1}{n} \cdot \mathbf{max}_i. \tag{8}$$

**Max Pooling** sets the first pooling weight $\psi_1$ to 1 and the others $\psi_i, i \neq 1$ to 0 to get pooling feature $\mathbf{x}_{mp} \in \mathcal{R}^{d_x}$,

$$\mathbf{x}_{mp} = \mathbf{max}_1. \tag{9}$$

However, the aforementioned two pooling approaches rely on fixed pooling strategies, posing a challenge in ensuring the effective retention of important information.

**Adjustable Multi-curvature Pooling** automatically adjusts pooling weights to obtain a superior pooling strategy, effectively retaining important information. To learn appropriate pooling weights $\Psi$ for the different positions of $\mathbf{M}$, i.e. $\mathbf{max}_i$, we first utilize the positional encoding strategy in [7, 31] to get positional encoding $\mathbf{P} = \{\mathbf{p}_1, \mathbf{p}_2, \cdots, \mathbf{p}_n\}$, $\mathbf{p}_i \in \mathcal{R}^{d_p}$. This positional encoding $\mathbf{P}$ contains prior information between position indices, and can be formulated as follows,

$$\mathbf{p}_i(2k) = sin(\frac{i}{10000^{2k/d_p}})$$
$$\mathbf{p}_i(2k+1) = cos(\frac{i}{10000^{2k/d_p}}), \tag{10}$$

where $k$ indicates the dimension. Then we regard the sequence of positional encoding $\mathbf{P}$ as input and utilize Bi-GRU [27] and Multi-Layer Perceptron (MLP) to obtain pooling weights $\Psi$,

$$\Psi = \text{MLP}(\text{Bi-GRU}(\mathbf{P})). \tag{11}$$

Further, $\Psi$ is normalized as follows,

$$\psi_i = \frac{\exp(\psi_i)}{\sum_{j=1}^{n} \exp(\psi_j)}. \tag{12}$$

Based on the learned pooling weights, we get the pooling feature $\mathbf{x}_{amp} \in \mathcal{R}^{d_x}$,

$$\mathbf{x}_{amp} = \sum_{i=1}^{n} \psi_i \cdot \mathbf{max}_i. \tag{13}$$

According to (10), (11), (12) and (13), the entire calculation process of AMP can be integrated as follows,

$$\mathbf{x}_{amp} = \text{AMP}(\mathbf{X}, \theta), \tag{14}$$

where $\theta$ indicates all the learnable parameters.

**Pooling Procedure.** We concatenate the space-shared and space-specific representations of the entity, relation and timestamp into $\mathbf{H} = \{\mathbf{h}_{\mathbb{S}}^{I}, \mathbf{h}_{\mathbb{H}}^{I}, \mathbf{h}_{\mathbb{E}}^{I}, \mathbf{h}_{\mathbb{S}}^{S}, \mathbf{h}_{\mathbb{H}}^{S}, \mathbf{h}_{\mathbb{E}}^{S}\}$ ($\mathbf{h} \in \{\mathbf{s}, \mathbf{r}, \mathbf{t}\}$). Subsequently, we employ the AMP approach to effectively retain important information among entities, relations, and timestamps, and the score function can be defined as follows,

$$f(\mathbf{s}, \mathbf{r}, \mathbf{o}, \mathbf{t}) = \langle \text{AMP}(\mathbf{H}, \theta), \mathbf{o} \rangle, \tag{15}$$

where $\langle \cdot, \cdot \rangle$ represents the inner product operation.

### 4.4 Loss Function

In this section, we propose the overall loss of the proposed model IME as follows,

$$\mathcal{L} = \mathcal{L}_{task} + \alpha \mathcal{L}_{sim} + \beta \mathcal{L}_{diff} + \gamma \mathcal{L}_{stru}, \tag{16}$$

where $\alpha, \beta, \gamma$ are the hyper-parameters. Each component within the loss is responsible for achieving the desired properties.

**Task Loss.** Following the strategy in [38], we explore the cross-entropy and standard data augmentation protocol to achieve the multi-class task,

$$\mathcal{L}_{task} = -\log\left(\frac{\exp(f(\mathbf{s}, \mathbf{r}, \mathbf{o}, \mathbf{t}))}{\sum_{\mathbf{s}' \in \mathcal{E}} \exp(f(\mathbf{s}', \mathbf{r}, \mathbf{o}, \mathbf{t}))}\right)$$
$$-\log\left(\frac{\exp(f(\mathbf{o}, \mathbf{r}^{-1}, \mathbf{s}, \mathbf{t}))}{\sum_{\mathbf{o}' \in \mathcal{E}} \exp(f(\mathbf{o}', \mathbf{r}^{-1}, \mathbf{s}, \mathbf{t}))}\right). \tag{17}$$

**Similarity Loss.** The purpose of the similarity loss is to minimize the disparities among shared features across different curvature spaces, aiming to bridge spatial gaps among them. Specifically, Central Moment Discrepancy (CMD) [43] is a distance metric employed to evaluate the similarity between two distributions by quantifying the discrepancy in their central moments. A smaller CMD value indicates a higher similarity between the two distributions. Let $X$ and $Y$ be bounded independent and identically distributed random vectors from two probability distributions, $p$ and $q$, defined on the interval $[a, b]$. The CMD can be defined as,

$$\text{CMD}(X, Y) = \frac{1}{|b - a|} \parallel \mathbf{E}(X) - \mathbf{E}(Y) \parallel_2$$
$$+ \sum_{k=2}^{\infty} \frac{1}{|b - a|^k} \parallel c_k(X) - c_k(Y) \parallel_2, \tag{18}$$

where $\mathbf{E}(X)$ is the expectation of $X$, and $c_k(x) = \mathbf{E}((X - \mathbf{E}(X))^k)$ is the central moment vector of order $k$.

In our case, we calculate the similarity loss through CMD,

$$\mathcal{L}_{sim} = \frac{1}{3} \sum_{(\mathbb{M}_1, \mathbb{M}_2)} \text{CMD}(\mathbf{h}_{\mathbb{M}_1}^{S}, \mathbf{h}_{\mathbb{M}_2}^{S}) \tag{19}$$

where $(\mathbb{M}_1, \mathbb{M}_2) \in \{(\mathbb{E}, \mathbb{H}), (\mathbb{E}, \mathbb{S}), (\mathbb{H}, \mathbb{S})\}$.

Table 1: Statistic information of whole datasets.

| Datasets | #Entities | #Relations | #Timestamps | #Time Span | #Granularity | #Training | #Validation | #Test |
|---|---|---|---|---|---|---|---|---|
| ICEWS14 | 6,869 | 230 | 365 | 2014 | 1 day | 72,826 | 8,941 | 8,963 |
| ICEWS05-15 | 10,094 | 251 | 4,017 | 2005-2015 | 1 day | 368,962 | 46,275 | 46,092 |
| GDELT | 500 | 20 | 366 | 2015-2016 | 1 day | 2,735,685 | 341,961 | 341,961 |

**Difference Loss.** The difference loss is designed to capture characteristic features of different curvature spaces through a similarity function. Specifically, we not only impose the soft orthogonality constraint between the shared and specific features but also between the space-specific features. The difference loss is calculated as:

$$\mathcal{L}_{diff} = \sum_{\mathbb{M}} \parallel (\mathbf{h}_{\mathbb{M}}^{S})^{T} \mathbf{h}_{\mathbb{M}}^{I} \parallel_{F}^{2} + \sum_{(\mathbb{M}_1, \mathbb{M}_2)} \parallel (\mathbf{h}_{\mathbb{M}_1}^{S})^{T} \mathbf{h}_{\mathbb{M}_2}^{S} \parallel_{F}^{2}, \quad (20)$$

where $\mathbb{M} \in \{\mathbb{S}, \mathbb{H}, \mathbb{E}\}$, $(\mathbb{M}_1, \mathbb{M}_2) \in \{(\mathbb{E}, \mathbb{H}), (\mathbb{E}, \mathbb{S}), (\mathbb{H}, \mathbb{S})\}$, $\parallel \cdot \parallel_{F}^{2}$ is the squared Frobenius norm.

**Structure Loss.** The structure loss [11] aims to ensure the structural similarity of quadruplets across various curvature spaces. Specifically, we define the relation on a triplet of samples $(\mathbf{x}_a, \mathbf{x}_b, \mathbf{x}_c)$ as the following cosine value:

$$\cos \angle \mathbf{r}_a \mathbf{r}_b \mathbf{r}_c = \langle \mathbf{e}^{ab}, \mathbf{e}^{cb} \rangle \quad \text{where} \quad \mathbf{e}^{ij} = \frac{\mathbf{r}_i - \mathbf{r}_j}{\parallel \mathbf{r}_i - \mathbf{r}_j \parallel_2} \quad (21)$$

where $\mathbf{r}_*$ is sample. Thus, the structure loss can be calculated as,

$$\mathcal{L}_{stru} = \frac{1}{3} \sum_{(\mathbb{M}_1, \mathbb{M}_2)} \parallel \cos \angle \mathbb{M}_{1_a} \mathbb{M}_{1_b} \mathbb{M}_{1_c} - \cos \angle \mathbb{M}_{2_a} \mathbb{M}_{2_b} \mathbb{M}_{2_c} \parallel_1,$$
$$(22)$$

where $(\mathbb{M}_1, \mathbb{M}_2) \in \{(\mathbb{E}, \mathbb{H}), (\mathbb{E}, \mathbb{S}), (\mathbb{H}, \mathbb{S})\}$.

## 5 EXPERIMENT

In this section, we provide detailed information about the datasets, describe the experimental setups, present experimental results, and conduct a comprehensive analysis of experimental results.

### 5.1 Datasets

We provide a list of three commonly-used TKG datasets and their key statistics are summarized in TABLE 1. **ICEWS14** and **ICEWS05-15** [12] are subsets of *Integrated Crisis Early Warning System (ICEWS)*, which encompass various political events along with their respective timestamps. **GDELT** [22] is a subset of the larger *Global Database of Events, Language, and Tone (GDELT)* that includes data on human social relationships.

### 5.2 Baselines

The proposed model is compared with some classic KGC methods, including SKGC and TKGC methods.

*A. SKGC methods:*

- **TransE** [3]: TransE introduces the translation mechanism into the knowledge graph completion (KGC) task;
- **DistMult** [41]: DistMult is a classic semantic matching-based KGC methods;

- **SimplE** [18]: SimplE develops a CP-based tensor decomposition method to ensure the independence between two embedding vectors of an entity;
- **RotatE** [29]: RotatE regards relation as a rotation between the head entity and the tail entity.

*B. TKGC methods:*

- **TA-DistMult** [12]: TA-DistMult applies RNN to associate the relation with the timestamp;
- **TeRo** [40]: TeRo first models the translation mechanism in the complex space;
- **DE-SimplE** [13]: DE-SimplE applies the diachronic entity embedding function to reflect the evolution of entities over timestamps;
- **ATiSE** [39]: ATiSE uses additive time series decomposition to model the temporal knowledge graph;
- **ChronoR** [25]: ChronoR takes the timestamp-relation pair as the rotation between the head entity and the tail entity;
- **TeLM** [38]: TeLM enhances the generalization capacity by the multi-vector embeddings of the entity in TKGC;
- **TuckERTNT** [28]: TuckERTNT extends TuckER [2] from the SKGC to the TKGC;
- **BoxTE** [24]: BoxTE extends BoxE [1] with the dedicated time embeddings;
- **BDME** [42]: BDME models TKGs with the multi-granularity embeddings;
- **EvoExplore** [44]: EvoExplore understands the evolutionary nature of TKGs by capturing local and global structures;
- **DyERNIE** [16]: DyERNIE first models TKGs within multi-curvature spaces;
- **BiQCap** [45]: BiQCap combines Euclidean and hyperbolic embeddings for each relation to explore complex patterns;
- **QDN** [34]: QDN designs a quadruplet distributor to facilitate the information interaction among entities, relations, and timestamps.

### 5.3 Link Prediction Metrics

We substitute either the head or tail entity in each test quadruplet $(\mathbf{s}, \mathbf{r}, \mathbf{o}, \mathbf{t})$ with all feasible entities sampled from the TKG. Subsequently, we rank the scores calculated by the score function. We employ Mean Reciprocal Rank (MRR) and Hit@$N$ as evaluation metrics, with $N$=1, 3 and 10. Higher values indicate better performance. Finally, we present the *filtered* results as final experimental results, which exclude all corrupted quadruplets from the TKG.

### 5.4 Parameters Setting

We use a grid search to find the best hyper-parameters based on the MRR performance on the validation dataset. Specifically, we

**Table 2: Link prediction results on ICEWS14, ICEWS05-15, and GDELT datasets. The best results are in bold and the second results are underlined. - means the result is unavailable.**

| Datasets | ICEWS14 | | | | ICEWS05-15 | | | | GDELT | | | |
|---|---|---|---|---|---|---|---|---|---|---|---|---|
| Metrics | MRR | Hit@1 | Hit@3 | Hit@10 | MRR | Hit@1 | Hit@3 | Hit@10 | MRR | Hit@1 | Hit@3 | Hit@10 |
| TransE (2013) | 0.280 | 0.094 | – | 0.637 | 0.294 | 0.090 | – | 0.663 | 0.113 | 0.0 | 0.158 | 0.312 |
| DistMult (2015) | 0.439 | 0.323 | – | 0.672 | 0.456 | 0.337 | – | 0.691 | 0.196 | 0.117 | 0.208 | 0.348 |
| SimplE (2018) | 0.458 | 0.341 | 0.516 | 0.687 | 0.478 | 0.359 | 0.539 | 0.708 | 0.206 | 0.124 | 0.220 | 0.366 |
| RotatE (2019) | 0.418 | 0.291 | 0.478 | 0.690 | 0.304 | 0.164 | 0.355 | 0.595 | – | – | – | – |
| TA-DistMult (2018) | 0.477 | 0.363 | – | 0.686 | 0.474 | 0.346 | – | 0.728 | 0.206 | 0.124 | 0.219 | 0.365 |
| ATiSE (2019) | 0.550 | 0.436 | 0.629 | 0.750 | 0.519 | 0.378 | 0.606 | 0.794 | – | – | – | – |
| DE-SimplE (2020) | 0.526 | 0.418 | 0.592 | 0.725 | 0.513 | 0.392 | 0.578 | 0.748 | 0.230 | 0.141 | 0.248 | 0.403 |
| TeRo (2020) | 0.562 | 0.468 | 0.621 | 0.732 | 0.586 | 0.469 | 0.668 | 0.795 | 0.245 | 0.154 | 0.264 | 0.420 |
| ChronoR (2021) | 0.625 | 0.547 | 0.669 | 0.773 | 0.675 | 0.596 | 0.723 | 0.820 | – | – | – | – |
| TeLM (2021) | 0.625 | 0.545 | 0.673 | 0.774 | 0.678 | 0.599 | 0.728 | 0.823 | – | – | – | – |
| TuckERTNT (2022) | 0.604 | 0.521 | 0.655 | 0.753 | 0.638 | 0.559 | 0.686 | 0.783 | 0.381 | 0.283 | 0.418 | 0.576 |
| BoxTE (2022) | 0.613 | 0.528 | 0.664 | 0.763 | 0.667 | 0.582 | 0.719 | 0.820 | 0.352 | 0.269 | 0.377 | 0.511 |
| EvoExplore (2022) | 0.725 | 0.653 | 0.778 | 0.852 | 0.790 | 0.719 | **0.843** | **0.915** | 0.514 | 0.353 | 0.602 | 0.748 |
| BDME (2023) | 0.635 | 0.555 | 0.683 | 0.778 | – | – | – | – | 0.278 | 0.191 | 0.299 | 0.448 |
| QDN (2023) | 0.643 | 0.567 | 0.688 | 0.784 | 0.692 | 0.611 | 0.743 | 0.838 | 0.545 | 0.481 | 0.576 | 0.668 |
| DyERNIE (2020) | 0.669 | 0.599 | 0.714 | 0.797 | 0.739 | 0.679 | 0.773 | 0.855 | 0.457 | 0.390 | 0.479 | 0.589 |
| BiQCap (2023) | 0.643 | 0.563 | 0.687 | 0.798 | 0.691 | 0.621 | 0.738 | 0.837 | 0.273 | 0.183 | 0.308 | 0.469 |
| IME | **0.819** | **0.790** | **0.835** | **0.872** | **0.796** | **0.750** | 0.821 | 0.875 | **0.624** | **0.485** | **0.754** | **0.791** |

tune the similarity loss weight $\alpha$, the difference loss weight $\beta$, and the structure loss weight $\gamma$, choosing from $\{0.1, 0.2, \cdots, 0.9\}$. The optimal $\alpha$, $\beta$ and $\gamma$ on different datasets are set as follows: $\alpha = 0.4$, $\beta = 0.4$ and $\gamma = 0.1$ for ICEWS14; $\alpha = 0.9$, $\beta = 0.3$ and $\gamma = 0.1$ for ICEWS05-15; $\alpha = 1$, $\beta = 0.3$ and $\gamma = 0.1$ for GDELT. We set the optimal embedding dimension $D$ to 500 across all datasets. For the AMP approach, the dimension of positional encoding is set to 32, i.e., $d_p$ is set to 32. The dimension of Bi-GRU is also set to 32 and MLP is used to project features from 32 dimensions to 1.

Moreover, the learning rate is fine-tuned within the range $\{0.1, 0.05, 0.01, 0.005, 0.001\}$ on different datasets, ultimately being set to 0.1 for all datasets. The batch size of 1000 is consistently applied across all datasets. The entire experiment is implemented using the PyTorch 1.8.1 platform and conducted on a single NVIDIA RTX A6000 GPU.

## 5.5 Experimental Results and Analysis

The link prediction experimental results are displayed in TABLE 2, and the experimental analyses are listed as follows:

(1) The proposed model outperforms state-of-the-art baselines on three datasets, showing clear superiority in most metrics. For example, the proposed model obtains 9.4% and 0.6% improvements over EvoExplore under MRR on ICEWS14 and ICEWS05-15, respectively. This phenomenon indicates that a single space is insufficient for modeling complex geometric structures concurrently, and the spatial gap in multi-curvature spaces severely limits the expressive capacity of TKGC models.

(2) BiQCap [45] and DyERNIE [16] are two important baselines because they both model TKGs in multi-curvature spaces. However,

our proposed method still improves most metrics for all datasets. This phenomenon reflects that our proposed method can effectively reduce spatial gaps caused by the heterogeneity of different curvature spaces.

(3) QDN [34] is also an essential baseline because it serves as a key component of the multi-curvature embeddings module. When compared to QDN, our proposed method exhibits a substantial improvement in performance across all metrics. This observation underscores the inadequacy of a single Euclidean space for modeling complex geometric structures.

These observations indicate that our proposed method can not only model complex geometric structures but also effectively reduce spatial gaps among different curvature spaces.

## 5.6 Impact of Loss Weights $\alpha$, $\beta$, and $\gamma$

In this experiment, we explore the influence of changing the loss weights $\alpha$, $\beta$, and $\gamma$ on MRR. As depicted in Figure 4, it becomes evident that with increasing weight, various loss functions display noteworthy differences in performance. To be specific, the similarity loss $\alpha$ and the difference loss $\beta$ display a parabolic shape, with their peaks occurring at 0.4. In contrast, the structure loss $\gamma$ reveals an overall declining trend, gradually diminishing as the weight increases.

These phenomena clearly illustrate that appropriate weights for similarity and difference losses effectively facilitate the learning of common and characteristic features of entities, relations, and timestamps across multiple curvature spaces. Conversely, a higher weight for the structure loss restricts their flexibility in embeddings across these multiple curvature spaces.

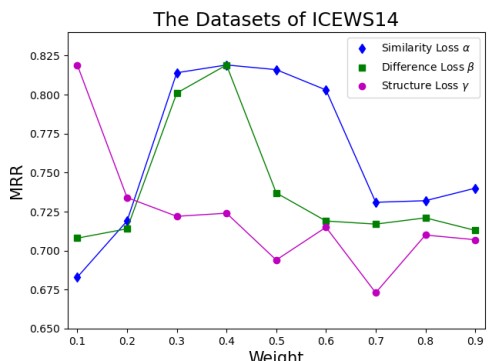

Figure 4: H@1 with varying loss weights $\alpha$, $\beta$ and $\gamma$ on ICEWS14.

Table 3: The ablation experiment results on ICEWS14. "w/o" represents removal for the mentioned factors, "(-)" denotes replacing Adjustable Multi-curvature Pooling (AMP) with the mentioned factors. We mark the better results in bolded font.

| Datasets | ICEWS14 | | | |
|---|---|---|---|---|
| Metrics | MRR | Hit@1 | Hit@3 | Hit@10 |
| DistMult (2015) | 0.439 | 0.323 | – | 0.672 |
| TA-DistMult (2018) | 0.477 | 0.363 | – | 0.686 |
| IME (-) MP | 0.523 | 0.430 | 0.574 | 0.696 |
| IME w/o $\mathcal{L}_{sim}$ | 0.740 | 0.693 | 0.765 | 0.824 |
| IME w/o $\mathcal{L}_{diff}$ | 0.716 | 0.653 | 0.752 | 0.835 |
| IME w/o $\mathcal{L}_{stru}$ | 0.810 | 0.760 | 0.810 | 0.859 |
| IME | **0.819** | **0.790** | **0.835** | **0.872** |

[1] **MP** represents *Max Pooling*.

## 5.7 Ablation Experiments

In order to investigate the impact of key modules and loss functions on performance, we conducted a series of ablation experiments, and the corresponding link prediction results are presented in TABLE 3.

i. "IME w/o $\mathcal{L}_{sim}$", "IME w/o $\mathcal{L}_{diff}$", and "IME w/o $\mathcal{L}_{stru}$" mean removing the similarity loss $\mathcal{L}_{sim}$, difference loss $\mathcal{L}_{diff}$, and structure loss $\mathcal{L}_{stru}$;

ii. "IME (-) MP" represents replacing Adjustable Multi-curvature Pooling (AMP) with the Max Pooling (MP).

(1) In the first category of ablation experiments, the proposed model achieves a significant improvement on ICEWS14. For example, compared to "IME w/o $\mathcal{L}_{sim}$", "IME w/o $\mathcal{L}_{diff}$", and "IME w/o $\mathcal{L}_{stru}$", the proposed model achieves 7.9%, 10.3%, and 0.9% improvements on Hit@1, respectively. Thus, we can summarize the following conclusions:

a) Similarity loss can effectively learn the commonalities across distinct curvature spaces and mitigate spatial gaps among them;

b) Difference loss can capture characteristic features specific to each space;

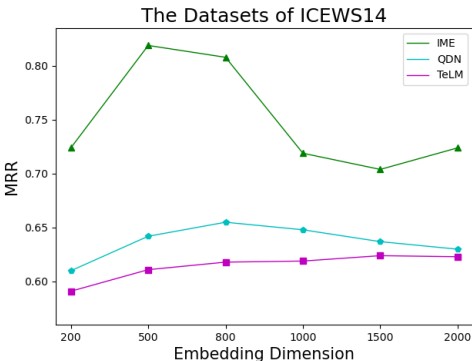

Figure 5: Comparison of MRR performance with different embedding dimensions on ICEWS14.

c) Structure loss serves to constrain the embeddings of entities, relations, and timestamps by ensuring that information in distinct spaces exhibits comparable geometric structures.

(2) In the second category of ablation experiments, the proposed model exhibits a certain improvement on ICEWS14. This phenomenon demonstrates that the adjustable multi-curvature pooling approach can effectively strengthen the important information for modeling the current TKG while weakening the undesirable ones.

## 5.8 Impact of Embedding Dimensions $D$

To empirically investigate the impact of embedding dimensions on ICEWS14, we fine-tune the dimension $D$ within the range of {200, 500, 800, 1000, 1500, 2000} and analyze the experimental results. As shown in Figure 5, the MRR performance on ICEWS14 exhibits an initial increase followed by a decrease as the dimension increases, eventually peaking at $D = 500$.

This phenomenon implies that the proposed model faces challenges in capturing intricate data relationships at lower dimensions, resulting in poorer performance. As the dimension increases, the model becomes more capable of effectively representing data, leading to enhanced performance. Nevertheless, beyond a certain threshold, this may introduce some issues such as overfitting or heightened complexity, consequently causing a decline in performance.

## 6 CONCLUSION

In this paper, we proposed a novel TKGC method called Integrating Multi-curvature shared and specific Embedding (IME). Specifically, IME models TKGs in multi-curvature spaces to capture complex geometric structures. Meanwhile, IME learns the space-specific property to comprehensively capture characteristic information, and the space-shared property to reduce spatial gaps caused by the heterogeneity of different curvature spaces. Furthermore, IME innovatively proposes an Adjustable Multi-curvature Pooling (AMP) approach to effectively strengthen the retention of important information. Experimental results on several well-established datasets incontrovertibly show that IME achieves competitive performance when compared to state-of-the-art TKGC methods.

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
