# OpenReview forum: "IME: Integrating Multi-curvature Shared and Specific Embedding for Temporal Knowledge Graph Completion"
_ACM.org/TheWebConf/2024/Conference — TheWebConf24 Oral_

### Official Review · Reviewer_jSZU · 2023-11-16

**Novelty:** 5
**Technical Quality:** 5

**Review:**

Summary:
This paper proposed a method which learns two key properties, namely space-shared property and space-specific property, to model TKGs in multi-curvature spaces to capture complex geometric structures. This paper proposes an adjustable multi-curvature pooling module, designed to attain a superior pooling strategy through training for the effective retention of important informationThe method proposed by this paper achieves competitive performance when compared to soat TKGC methods.

Paper Strength:
The proposed model in this paper outperforms soat baselines on three datasets. The AMP approach effectively strengthen the retention of important information.

Paper Weakness:
The paper lacks source code.

**Questions:**

Will the method proposed by this paper bring about a time loss that is difficult to ignore?

**Ethics Review Description:**

No.

**Reviewer Confidence:**

2: The reviewer is willing to defend the evaluation, but it is likely that the reviewer did not understand parts of the paper

**Scope:**

3: The work is somewhat relevant to the Web and to the track, and is of narrow interest to a sub-community

---

### Official Review · Reviewer_UzP2 · 2023-11-21

**Novelty:** 5
**Technical Quality:** 5

**Review:**

Quality: the quality of the paper is in wide parts high, with a focus on the experimental evaluation. Some more theoretical properties of the model would be interesting to understand

Clarity: is also very good throughout, with some polish of the text and formatting (e.g., in the earlier parts of section 5) being possible

Originality: the approach appears to be original, at least in a combination I have not seen so far - of course with the components being well known

Significance: the experimental evaluation shows significantly improved results, especially for ICEWS14 compared to SOTA

**Questions:**

1: The improvements are much more pronounced for ICEWS14 compared to ICEWS05-15. Is there some reason you can give or suppose for this fact?

2: The improvement of MRR for ICEWS14 is singificant, with a jump of almost 0.1. It is clear what the general improvements of the model are, but where do you see the concrete reason for this very large improvement relative to EvoExplore?

3: A somewhat less important question out of interest: the SKGC baselines - how did you specifically select them? (I do not doubt them)

**Reviewer Confidence:**

3: The reviewer is confident but not certain that the evaluation is correct

**Scope:**

3: The work is somewhat relevant to the Web and to the track, and is of narrow interest to a sub-community

---

### Official Review · Reviewer_oBKy · 2023-11-23

**Novelty:** 5
**Technical Quality:** 5

**Review:**

# Summary

The paper utilizes a Quadruplet Distributer Mechanism (QDM) introduced in QDN [34] to encode different spaces. However, QDN does not model different Space Curvatures (SC). For IME to understand which part of the different SCs are important, they apply two gating mechanisms, one with shared weights across SCs and one with a unique weight for each SC. All space embeddings for all $h\in\{s,r,t\}$ are aggregated using a novel weighted sum. The weight is learned using positional information only as each dimension is sorted. The gating mechanism and weighted sum are the novelties of the work.

The paper is generally well-written and of a high quality.

# Strengths

S1. Comparing to a large set of baselines, including very recent methods.

S2. Generally easy to follow the structure of the method and well written.

S3. Intuitive figures and tables improve the readability of the paper.

# Weaknesses

W1. No theoretical analysis of the method's capabilities or complexities.

W2. Limited analysis of novel pooling method.

W3. Why different part of the method is utilized is not always clear.

**Questions:**

# Questions and remarks

Q1. (W1) I wonder which kinds of patterns IME is able to capture compared to baselines. E.g., the well-known example of TransE not capturing symmetric relations and 1-to-N relations.

Q2. (W1) The method seems quite complex compared to other TKGC methods. It would be interesting to know the complexity compared to similar methods, such as QDN, EvoExplore, and BiQCap.

Q3. (W1) How the QDM should be computed for different spaces is not explained. As hyperbolic/hyperspherical vector operations are not known by all readers, there should at least be references to works explaining such computations. Ideally, these should be included as preliminaries for the work to be self-contained.

Q4. (W2) As the novelty is primarily in the pooling method. I would expect testing against at least AP and MP, but also well-known attention mechanisms such as self-attention. Furthermore, these experiments should be on all the datasets before it is possible to draw any conclusions.

Q5. (W3) It is clear how AMP is utilized, but why, e.g., the dimensions are sorted is not intuitively explained.

Q6. Inconsistent use of $\check{}$, e.g., between eq. 5 and 6 and between eq. 14 and 15.

Q7. There should be references to the papers where the results are from. Ideally, all results should be recomputed and validated; however, within TKGC/SKGC, reusing results seems to be the norm.

Q8. YAGO11k is often used with sparse links used by your baseline QDN. I am wondering why this work doesn't compare to it.

**Reviewer Confidence:**

3: The reviewer is confident but not certain that the evaluation is correct

**Scope:**

4: The work is relevant to the Web and to the track, and is of broad interest to the community

---

### Official Review · Reviewer_GX4H · 2023-11-27

**Novelty:** 4
**Technical Quality:** 5

**Review:**

# Overall

This work proposes a method for modeling temporal knowledge graphs which combines embeddings on multiple geometric spaces. Different pooling methods are proposed to combine the scores given by each of the embeddings. The combined method outperforms the existing methods, and the models that use a single geometric space.

Regarding the related work. I think that there is not enough revision on methods that combine different methods and geometries. The following works are missing:

- Gu, A., Sala, F., Gunel, B., Ré, C.: Learning mixed-curvature representations in product spaces. In: International Conference on Learning Representations (2018).

- Gregucci, C., Nayyeri, M., Hernández, D., Staab, S.: Link prediction with attention applied on multiple knowledge graph embedding models. In: ACM WebConf (2023).

- Wang, Y., Gemulla, R., Li, H.: On multi-relational link prediction with bilinear models. In: AAAI (2018)

- Wang, Y., Chen, Y., Zhang, Z., Wang, T.: A probabilistic ensemble approach for knowledge graph embedding. Neurocomputing (2022)

In general, the work is easy to read, but there are some issues with the writing. Next are some of them.

# Issues

**Line 16**. It is said that a TKGC methods have a capacity to *capture* intricate geometric structures. This wording is confusing. Such a method does not capture a geometry of the knowledge graph, but models the knowledge graph using a geometric representation. The geometries do not exist on the knowledge graph. The same issue is in lines 124-125.

**Line 46**. It is said that a directed edge is a relation. However, a relation is a set of tuples. An edge can represent a tuple in a relation, but it is not a relation. This wrong terminology is unfortunately getting common nowadays. The same issue appears in line 342.

**Line 341**. A graph $\mathcal{G}$ is notated as a set $\lbrace \mathcal{Q} \mid \mathcal{E}, \mathcal{R}, \mathcal{T} \rbrace$ defined by intension. This is a wrong notation for sets because the expression after the $|$ must be a condition. It would be better if it is notated as $(\mathcal{E}, \mathcal{R}, \mathcal{T}, \mathcal{Q})$, as usual.

**Lines 357-358**. It is said that a ring, a hierarchical, and a chain structure are geometrical structures. It is not clear to me what the authors mean by these structures, but I guess that a chain is a sequence of edges where the target node of one edge is the source node of the next edge. I am not sure what is the meaning of *geometric* here because the examples are not embedded into a geometric space.

**Line 384.** I guess that $\oplus$ is the element-wise sum, since it is already said that $\odot$ is an element-wise multiplication. It would be better to make this explicit.

**Line 352.** At the end of the line appears TABLE, and after a line break, it appears 3. Instead, it should be written Table~3 to avoid the line break.

**Questions:**

To confirm my understanding of the method. Which of the next two sentences is true?
1. There are three models, each one using a different geometric space, which are trained separately, and then these models are used to train a pooling system.
2. The three models are trained together with the pooling system.

**Ethics Review Description:**

No ethics review flag

**Reviewer Confidence:**

1: The reviewer's evaluation is an educated guess

**Scope:**

3: The work is somewhat relevant to the Web and to the track, and is of narrow interest to a sub-community

---

### Official Review · Reviewer_AUxy · 2023-11-30

**Novelty:** 6
**Technical Quality:** 6

**Review:**

Overall, the research of this paper is concerned with KG reasoning / completion (KGC).
Triple or (here) quadruplet queries are to be completed with the subject/object location missing.
More specifically, the research is concerned with Temporal KGC.
That is, an additional time property is involved.
Even more specifically, the research cares about the achievable performance, when exploiting the (multi-) curvature of (T)KGs.
The idea is to allow for a multi-curvature in a flexible manner.
Hyperspherical, hyperbolic, and Euclidean space are considered and combined to have these play out their strengths.
The approach (Integrating Multi-curvature shared and specific Embedding (IME)) emphasizes effective support/distinction of the space-shared versus -specific property.

The paper appears to provide (for this reviewer being an outsider anyway) a highly structured and in-depth discussion of related work.
The methodology section derives IME from previous work.
One can spot how the different spaces and their property of being space-specific or -shared affects, for example, the pooling.
The experiment identifies several types of well-argued baselines — also linked well to related work.
The experiment’s execution is quite intricate (for this reviewer being an outsider anyway), but it is easy to infer that the new method outperforms the baselines.

**Questions:**

The following questions are stated from the perspective of an outsider with little background in KGC.
Still it is hoped that the questions would also help other readers.
Why does the caption of Figure 2 talk about a different example than the actual illustration (Donald Trump versus Bill Gates)?
How do section 1-3 prepare for handling of “uncertainty and diversity inherent in knowledge”, as discussed in l208-209?
Would the early part of the paper be able to motivate hyperspherical, hyperbolic, and Euclidean spaces in how they capture some key aspects of TKG(C)? Could this be even hinted at in Figure 2?
What is the variation of the time dimension of time in TKGs? That is, does it matter how properties evolve over time and how is this different from using definite timestamps (e.g., for undisputed birth dates)? (Is this related to the distinction of space-shared and space-specific properties? Could Figure 2 do a better job in sorting this out?)

**Reviewer Confidence:**

1: The reviewer's evaluation is an educated guess

**Scope:**

3: The work is somewhat relevant to the Web and to the track, and is of narrow interest to a sub-community

---

### Decision · Program_Chairs · 2024-01-22

**Decision:**

Accept (Oral)

**Comment:**

The reviewers note the following strengths relating to the paper:

 * The paper is clear.
 * Detailed experimentation with many recent baselines.
 * Originality of the approach.
 * Very encouraging results.

 Some weaknesses noted by the reviewers:

 * Some issues with formal notation and writing.
 * Lack of theoretical analysis.
 * Rationale behind the design not always clear.
 * No source code provided.

 The authors' response led to discussions regarding these weaknesses, and to some reviewers increasing their scores.

 Overall the relevance, technical quality and novelty scores lean above average (with relevance being on the lower end).

 Based on the reviewer feedback, author response, scores, etc., I find this to be a relatively strong submission, and do not find any key reason to reject. Hence I recommend an Accept (though scores are only above average, it is notable that they are consistently so).